# Racial Distribution of Neighborhood-Level Social Deprivation in a Retrospective Cohort of Prostate Cancer Survivors

**DOI:** 10.3390/diseases10040075

**Published:** 2022-10-03

**Authors:** Oluwole Adeyemi Babatunde, John L. Pearce, Melanie S. Jefferson, Lewis J. Frey, Peggi M. Angel, Richard R. Drake, Caitlin G. Allen, Michael B. Lilly, Stephen J. Savage, Chanita Hughes Halbert

**Affiliations:** 1Department of Psychiatry and Behavioral Sciences, Medical University of South Carolina, Charleston, SC 29425, USA; 2Hollings Cancer Center, Medical University of South Carolina, Charleston, SC 29425, USA; 3Department of Public Health Sciences, Medical University of South Carolina, Charleston, SC 29425, USA; 4Department of Cell and Molecular Pharmacology and Experimental Therapeutic, Medical University of South Carolina, Charleston, SC 29425, USA; 5Department of Behavioral Sciences and Health Education, Rollins School of Public Health, Emory University, Atlanta, GA 30322, USA; 6Department of Medicine, Medical University of South Carolina, Charleston, SC 29425, USA; 7Department of Urology, Medical University of South Carolina, Charleston, SC 29425, USA; 8Department of Population and Public Health Sciences, University of Southern California, Los Angeles, CA 90032, USA

**Keywords:** prostate cancer, Social Deprivation Index, racial disparity

## Abstract

Background: A better understanding of neighborhood-level factors’ contribution is needed in order to increase the precision of cancer control interventions that target geographic determinants of cancer health disparities. This study characterized the distribution of neighborhood deprivation in a racially diverse cohort of prostate cancer survivors. Methods: A retrospective cohort of 253 prostate cancer patients who were treated with radical prostatectomy from 2011 to 2019 was established at the Medical University of South Carolina. Individual-level data on clinical variables (e.g., stage, grade) and race were abstracted. Social Deprivation Index (SDI) and Healthcare Professional Shortage (HPS) status was obtained from the Robert Graham Center and assigned to participants based on their residential census tract. Data were analyzed with descriptive statistics and multivariable logistic regression. Results: The cohort of 253 men consisted of 168 white, 81 African American, 1 Hispanic and 3 multiracial men. Approximately 49% of 249 men lived in areas with high SDI (e.g., SDI score of 48 to 98). The mean for SDI was 44.5 (+27.4), and the range was 97 (1–98) for all study participants. African American men had a significantly greater likelihood of living in a socially deprived neighborhood compared to white men (OR = 3.7, 95% C.I. 2.1–6.7, *p* < 0.01), while men who lived in areas with higher HPS shortage status were significantly more likely to live in a neighborhood that had high SDI compared to men who lived in areas with lower HPS shortages (OR = 4.7, 95% C.I. = 2.1–10.7, *p* < 0.01). African Americans had a higher likelihood of developing biochemical reoccurrence (OR = 3.7, 95% C.I. = 1.7–8.0) compared with white men. There were no significant association between SDI and clinical characteristics of prostate cancer. Conclusions: This study demonstrates that SDI varies considerably by race among men with prostate cancer treated with radical prostatectomy. Using SDI to understand the social environment could be -particularly useful as part of precision medicine and precision public health approaches and could be used by cancer centers, public health providers, and other health care specialists to inform operational decisions about how to target health promotion and disease prevention efforts in catchment areas and patient populations.

## 1. Introduction

Census-level neighborhood characteristics and self-reported socioeconomic factors (e.g., education level) interacted synergistically in terms of the risk of biochemical recurrence among African American and white prostate cancer patients [1]. African American and white prostate cancer patients who lived in neighborhoods with greater economic deprivation (e.g., low education, high poverty) had more aggressive disease, but the association between neighborhood deprivation and disease severity was most pronounced among African American men [1].

Social deprivation is often quantified using a composite variable that captures area-level deprivation based on social and economic characteristics including income, education, housing, household characteristics, transportation, percent racial minority, and unemployment. These characteristics are collected as part of the American Community Survey (https://www.census.gov/programs-surveys/acs/, accessed on 3 October 2020) and are often analyzed as a composite measure such as the Robert Graham Center’s Social Deprivation Index (SDI) [2]. Specifically, SDI is used to measure disadvantage across small geographic areas and is one strategy for characterizing social factors that are important to health care and clinical outcomes [2,3]. SDI is conceptually similar to neighbourhood deprivation, as it uses a similar analytical strategy to construct the index based on seven social and economic indicators [2]. In international samples and in the US, individuals who live in neighborhoods that have high levels of social deprivation have higher rates of cancer mortality [1,4,5,6]. In the US, social deprivation is characterized by factors such as low income and high levels of unemployment; these neighborhood characteristics mirror racial differences in socioeconomic variables. That is, African American people tend to have lower incomes and are more likely to be unemployed compared to white people [7]. Consistent with this, racial disparities in social deprivation have been found in prospective research with African American prostate cancer patients [8]. However, there may be less variation in social deprivation between African American people and white people in geographic areas that have other geographic risk factors for poor health care and outcomes [9]. South Carolina, for instance, is a predominantly rural state that has a large population of African Americans and other medically underserved groups who have reduced access to health care because of rurality and limited economic resources.

Knowing social deprivation is important for precision medicine approaches because it provides insight into the neighborhood-level factors that could be targeted as part of cancer control interventions. According to the definition by the National Institute of Health (NIH), precision medicine refers to a new treatment and prevention method based on understanding of individual gene, environment, and lifestyle [10]. Geographic variables are important in cancer outcomes among cancer survivors, as geographic inequalities indicate a potential for improving cancer care and survival [11]. Despite this, however, empirical data on the distribution of social deprivation among men who have been diagnosed with prostate cancer in a medically underserved state that has significant rurality and other geographic determinants of morbidity and mortality are not available. In this study, we examined the distribution of social deprivation in a racially diverse sample of prostate cancer survivors in a medically underserved state based on their clinical characteristics (e.g., Gleason score, tumor stage) and racial background.

## 2. Methods

*Study Population.* A retrospective cohort of prostate cancer patients who were treated with radical prostatectomy from 2011 to 2019 was established using the tumor registry at the Hollings Cancer Center at the Medical University of South Carolina. Data on clinical variables (e.g., stage, grade) and race were abstracted from electronic medical records. A retrospective cohort of 253 prostate cancer patients were included in this study. Of the 253 participants in our sample, 168 were White, 81 were African American, 1 was Hispanic and 3 were more than 1 race. The study excluded the 1 Hispanic and 3 that identified as multiple races because adding these patients in will result in very small groups that will reduce the power of any conclusions.

*Measures.* Individual-level determinants analyzed in this study included race, age, and marital status. These variables were abstracted from the electronic health record. Clinical factors included prostatic specific antigen at diagnosis, diagnosis-to-surgery days (determined by calculating number of days between date of diagnosis and date of surgery), stage of prostate cancer at the point of treatment (prostatectomy) diagnosis, grade/Gleason score of prostate cancer at stage of prostate cancer at the point of treatment (prostatectomy), and tumor aggression. Tumor stage and grade were categorized as low or high stage. Low stage was defined as stages 1 and 2 (localized disease: pT2, pT2a, pT2b and pT2c), and high-stage is defined as stages 3 and 4 (nonlocalized: pT3, pt3a, & pt3b). For tumor grade, low grade was defined as tumor Gleason score of 6 or below, Gleason score 7 is median grade, and high-grade is defined as a tumor score of 8 or greater [1]. Five comorbidities, namely, hypertension, heart problems, stroke, diabetes, and hyperlipidemia were summarized by how many each participant had to 0. 1, 2, 3, and 4. County-level Healthcare Professional Shortage (HPSA) status was extracted from Health Resources and Services Administration. HPSA values ranged from 1 to 26, with higher values representing greater health care professional shortage. The 25th and 75th percentile values were used to categorize patients into groups who were living in geographic areas with high versus low levels of health care professional shortage. Lower shortage (better) was categorized as HPSA scores 8–10, intermediate shortage was categorized as HPSA scores of 11–15, while higher deprivation (worse) was categorized as SDI scores 16–20.

Social deprivation was determined for study participants by first converting residential street addresses to spatial coordinates using a geocoder in ESRI’s ArcGIS geographic information systems (GIS) [2]. We next assigned 2010 census-tract-level identifiers to our study population using spatial overlay analyses that joined geocoded residential locations with and census boundary files from the US Census. We subsequently used our census tract identifier to link participant’s address to obtain census-tract-level SDI scores from the Robert Graham Center that were originally intended to provide a composite measure of health care access and need (Buter et al.) [3].

The SDI value derived from the Robert Graham Center calculations which was assigned to each of the participant in our cohort ranged from 1 to 100, with higher values reflecting greater deprivation. SDI was analyzed as both a continuous and dichotomous variable in this study to increase our understanding of the distribution of deprivation levels across the state. The 25th and 75th percentile were utilized to categorize the SDI into lower deprivation, intermediate deprivation, and higher deprivation areas. The median value was used to categorize patients into groups who were living in geographic areas with high versus low levels of social deprivation. Biochemical reoccurrence (yes/no) was determined by prostate specific antigen value ≥ 0.2 ng/mL following radical prostatectomy [12]. This study was reviewed and approved by the institutional review board at Medical University of South Carolina.

*Statistical Analyses*. Statistical analyses were performed utilizing SAS software version 9.4. First, descriptive statistics were generated to characterize the study sample in terms of racial background, clinical characteristics, and SDI. Next, Chi Square Tests of Association was used to examine the bivariate relationships between SDI and race, age, and clinical factors. Multivariate logistic regression analysis was conducted to identify factors having significant independent associations with SDI. Variables that had a *p* < 0.20 with SDI in the bivariate analysis was included in the regression model. Covariates with a *p* ≤ 20 that were added to the model is justified by previous study where this selection method helped to select the variables to fit the best model in subsequent multivariable logistic regression model [13]. In exploring the multivariable logistic regression determining SDI, the variables that were entered into the model are race, age, marital status and HPSA. In exploring the multivariable logistic regression between race, SDI and biochemical reoccurrence, the variables that were entered into the model are race and SDI with adjustments for healthcare professional shortage, age, marital status, prostate specific antigen, stage, grade/Gleason score.

## 3. Results

Table 1 shows the characteristics of the study sample. Sixty-seven percent of patients were white and 33% were African American. The mean (SD) age of patients was 67 years (±6.6) and 85% of patients were married. There were no racial differences in prostate cancer stage or grade, but white men were significantly more likely than African American men to be married (88.7% versus 77.8%, *p* = 0.02). The proportion of African American men that lived in areas with high healthcare professional shortage was higher (68%) compared with the proportion of white men (52%, *p* = 0.02).

The mean (SD) for SDI was 44.5 (±27.4) and the range was 97 (1–98). More than 80% of the counties in which men lived had high levels of social deprivation (not shown in Tables/Figures). Table 2 shows the bivariate analysis of social deprivation using the binary SDI variable of high versus low categories. There were significant racial differences the proportion of men who lived in areas with high SDI; 72% of African American men lived in neighborhoods with high deprivation compared to 38% of white men who lived in neighborhoods with high deprivation. Consistent with this, the mean SDI was 58.5 (SD) among African American men compared to 37.8 (SD) among white men (t = −2.1, *p* = <0.01). Table 3 shows the results of the multivariate logistic regression analysis of social deprivation. African American men had a significantly greater likelihood of living in a geographic area with high social deprivation compared to white men (OR = 3.7, 95% CI: 2.1–6.7). Additionally, the likelihood of living in a geographic area with high social deprivation was higher among men who lived in areas with higher healthcare professional shortage (OR: 2.1, 95% CI: 1.2–3.6). African Americans had a higher likelihood of developing biochemical reoccurrence (OR = 3.7, 95% C.I. = 1.7–8.0) compared with white men after adjustments were made for healthcare professional shortage, age, marital status, prostate specific antigen, stage, grade/Gleason score (Table 4). There were no statistically significant associations between SDI and biochemical reoccurrence.

## 4. Discussion

The purpose of this study was to characterize social deprivation among men who have a personal history of prostate cancer and to identify factors having significant independent associations with social deprivation. Geographic factors and residency in a particular geographic region have implications for the types of health care services that individuals are able to access; the policies that govern when and how cancer care services are obtained; and the resources that exist for health promotion, disease control, and cancer treatment. For instance, Vetterlein and colleagues (2017) found that prostate cancer patients who traveled at least 50 miles for treatment had lower overall mortality compared to those who traveled less than 12 miles [14]. This may be because patients who are able to travel greater distances for treatment have increased access to high volume academic medical centers where NCI-designated cancer centers are likely to be located. Additionally, previous studies have shown that African American populations usually reside close to inner-city academic medical centers, while white higher-socioeconomic populations usually lived in fringe areas that are more suburban [15,16]. Several studies have examined travel distance and cancer outcomes, [14,17] but limited empirical data are available on the distribution of social deprivation in cancer patients and survivors.

Social deprivation reflects the extent to which individuals in a community do not have access to services and other resources that are important to health care and outcomes; the percent of the population who does not have a car is one aspect of social deprivation [2]. Even though patients in this study were able to overcome potential transportation challenges that can reduce access to obtaining treatment at an NCI-designated cancer center, approximately 50% of patients in our study lived in a geographic area that had high levels of social deprivation. Shafique and colleagues found that living in a geographic area with high social deprivation is associated with an increased risk of cancer mortality, [18] but with the exception of racial background, none of the prognostic factors associated with prostate cancer mortality were related to social deprivation in the present study. This may be because of the limited variability in social deprivation in South Carolina (about to 80% of counties had high levels of social deprivation) and almost half of the men in this study lived in areas that have high social deprivation. Another possible explanation is that our sample only included prostate cancer patients who were treated with radical prostatectomy. Patients who have early stage or locally advanced disease are optimal candidates for radical prostatectomy based on their prognostic variables (e.g.,). Zeigler-Johnson and colleagues found that greater neighborhood deprivation was associated with adverse prognostic variables in a state-based sample of prostate cancer patients who had more diverse stages of disease and treatment [1]

We also found a significant association between healthcare professional shortage status and living in a geographic area that has high SDI. Previous studies have shown that a lack of specialists and primary care physicians at the county level was a potential cause of geographic variations for late-stage cancer risk and mortality [16,19,20]. The association between healthcare professional shortage and SDI in our sample underscores the importance of implementing interventions to reduce physician shortages in socially deprived neighborhoods [21]. Additional research is needed to identify effective strategies for reducing health care professional shortages; telehealth and telemedicine are among the efforts that may improve access to health care in geographic areas that have high levels of social deprivation [21,22]. One limitation of this study is that most patients at our institution were treated with radical prostatectomies; adding many types of radiation therapies, e.g., brachytherapy, photon beams, Image-Guided, Intensity-Modulated Radiation Therapy, or Image-Guided Radiation Therapy would result in very small treatment groups that will reduce the power of any conclusions.

Perhaps not surprisingly, African American patients were more likely than white patients to live in geographic areas with high social deprivation. Social deprivation is based on census-level socioeconomic characteristics that include income, education, and employment status; national data have shown that African American people have lower levels of these variables compared to white people [2]. Thus, the greater levels of social deprivation observed among African Americans and the increased likelihood of living in a socially deprived neighborhood among these men in the present study is consistent with racial differences in socioeconomic factors. Research is now being conducted to examine the effects of multilevel determinants of prostate cancer survival; this work has shown that both individual-level risk factors (race) and neighborhood-level variables are associated with survival following prostate cancer diagnosis [23,24]. Contextual factors related to neighborhood variables are important components of multilevel frameworks of minority health and health disparities [16]; however, census-level data may not be integrated into local tumor registries and data warehouses at academic and community oncology cancer centers. Further, basic scientists and clinical researchers may have limited training in multilevel statistical techniques that should be used to analyze individual- and neighborhood-level variables. The SDI is a composite score that could be integrated into local tumor registries, basic science and clinical studies, and population-based research to capture the effects of neighborhood-level variables. When added to tumor registry data, the SDI can provide insight into the social environment in which men are living. Using SDI to understand the social environment could be particularly useful as part of precision medicine and precision public health approaches and could be used by cancer centers, public health providers, and other health care specialists to inform operational decisions about how to target health promotion and disease prevention efforts in catchment areas and patient populations.

## Figures and Tables

**Table 1 diseases-10-00075-t001:** Sociodemographic and clinical characteristics (*n =* 249).

	Overall (*n =* 249) n (%)	White (*n =* 168) n (%)	African American (*n =* 81) n (%)	*p*-Value
Age, mean (SD)	67.3 (6.6)	67.8 (6.7)	66.3 (6.3)	0.09
Prostatic Specific Antigen (SD)	9.3 (11.1)	9.6 (11.9)	8.6 (9.2)	0.47
Diagnosis to Surgery days (SD)	111 (182)	110.1 (196.6)	113.4 (147.3)	0.88
Married				
No	37 (14.9)	19 (11.3)	18 (22.2)	0.02
Yes	212 (85.1)	149 (88.7)	63 (77.8)	
Stage				
Non-organ confined (High)	48 (19.3)	33 (19.6)	15 (19.0)	0.90
Organ confined (Low)	199 (79.9)	135 (80.4)	64 (81.0)	
Grade/Gleason score				0.94
High (Gleason 8 & 9)	10 (4.0)	7 (4.2)	3 (3.7)	
Median (Gleason 7)	8 (3.2)	5 (3.7)	3 (3.7)	
Low (Gleason 5 & 6)	231 (92.8)	156 (92.9)	75 (92.6)	
Composite SDI score, mean (SD)	44.5 (27.4)	37.8 (25.0)	58.5 (26.9)	<0.01
Composite SDI score				
Lower Deprivation	97 (39.0)	81 (48.2)	16 (19.8)	<0.01
Intermediate Deprivation	69 (27.7)	46 (27.4)	23 928.4)	
Higher Deprivation	83 (33.3)	41 (24.4)	42 (51.9)	
Healthcare professional shortage status (SD)	12.5 (5.0)	11.7 (5.2)	14.0 (4.0)	<0.01
Healthcare professional shortage status (SD)				
Lower shortage	100 (40.2)	76 (45.2)	24 (29.6)	<0.01
Intermediate shortage	99 (39.8)	67 (39.9)	32 (29.5)	
Higher shortage	50 (20.1)	25 (14.9)	25 (30.9)	
Number of Comorbidities				
0	114 (45.8)	83 (49.4)	31 (38.3)	0.22
1	62 (24.9)	43 (25.6)	19 (23.5)	
2	48 (19.3)	28 (16.7)	20 (24.7)	
3	22 (8.9)	13 (7.7)	9 (11.1)	
4	3 (1.2)	1 (0.6)	2 (2.5)	

**Table 2 diseases-10-00075-t002:** Association between sociodemographic and clinical characteristics and social deprivation (Chi Square analysis).

	Lower Deprivation n (%)	Intermediate Deprivation n (%)	Higher Deprivation n (%)	Chi Square	*p*-Value
**Characteristic**					
Race					
White men	81 (48.2)	46 (27.4)	41 (24.4)	23.7	<0.01
African American men	16 (19.8)	23 (28.4)	42 (51.9)		
Age (category)					
68–80 years old	60 (44.1)	32 (25.5)	44 (32.4)	4.0	0.13
47–67 years old	37 (32.7)	37 (32.7)	39 (34.5)		
Marital Status					
Married	83 (39.2)	62 (29.3)	67 (31,6)	2.5	0.29
Not Married	14 (37.8)	7 (18.9)	16 (43.2)		
Gleason score/Grade					
Low (5,6)	93 (40.3)	59 (25.5)	79 (34.2)	0.1	0.10
Median (7)	2 (25.0)	4 (50.0)	2 (25.0)		
High (8,9)	2 (20.0)	6 (60.0)	2 (20.0)		
Stage at Diagnosis					
Low	77 (38.7)	53 (26.6)	69 (34.7)	1.1	0.59
High	20 (41.7)	15 (31.3)	13 (27.1)		
Healthcare professional shortage status (SD)					
Lower shortage	53 (53.0)	20 (20.0)	27 (27.0)	27.7	<0.01
Intermediate shortage	38 (38.4)	33 (32.0)	28 (28.3)		
Higher shortage	6 (12.0)	16 (32.0)	28 (56.0)		
Number of Comorbidities					
0	43 (37.7)	35 (30.7)	36 (31.6)	5.3	0.72
1	26 (41.9)	18 (29.0)	18 (29.0)		
2	21 (43.8)	8 (16.7)	19 (40.0)		
3	6 (27.3)	7 (31.8)	9 (40.9)		
4	1 (33.3)	1 (33.3)	1 (33.3)		

**Table 3 diseases-10-00075-t003:** Multivariate logistic regression model of social deprivation.

	Multivariable Logistic Regression Analysis	
Characteristic	Higher Deprivation (SDI > 47) 121 (48.59) Number (%)	Odd Ratio (95% Confidence Interval)
Race		
White men	63 (37.5)	Reference
African American men	58 (71.6)	3.6 (2.0–6.6)
Age (category)		
68–80 years old	60 (44.1)	Reference
47–67 years old	61 (54.0)	1.0 (0.9–1.0)
Marital status at diagnosis		
Married	99 (46.7)	Reference
Not Married	22 (59.5)	0.9 (0.6–1.5)
Healthcare professional shortage status (SD)		
Lower shortage	36 (29.8)	Reference
Intermediate shortage	46 (38.0)	1.4 (0.8–2.6)
Higher shortage	39 (32.2)	4.8 (2.1–11.1)
Gleason score/Grade		
Low (5,6)	79 (34.2)	Reference
Median (7)	2 (25.0)	0.8 (0.2–3.7)
High (8,9)	2 (20.0)	1.4 (0.4–5.8)

**Table 4 diseases-10-00075-t004:** Multivariate logistic regression model of biochemical reoccurrence.

	^1^ Multivariable Logistic Regression Analysis	
Characteristic	Biochemical Reoccurrence (Yes) Number (%)	Odd Ratio (95% Confidence Interval)
Race		
White men	26 (50.0)	Reference
African American men	26 (50.0)	3.7 (1.7–8.0)
Social Deprivation Index		
Lower deprivation	18 (34.6)	Reference
Intermediate deprivation	13 (25.0)	0.8 (0.3–2.2)
Higher deprivation	21 (40.4)	0.9 (0.4–2.4)

^1^ Multivariable logistic regression model with race and social deprivation index as the main predictor variables for biochemical reoccurrence and adjustments made for healthcare professional shortage, age, marital status, prostate specific antigen, stage, grade/Gleason score.

## Data Availability

Not applicable.

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
