# Peer review of "Racial Distribution of Neighborhood-Level Social Deprivation in a Retrospective Cohort of Prostate Cancer Survivors"

_diseases, 2022, doi:10.3390/diseases10040075_

Round 1

Reviewer 1 Report (New Reviewer)

The manuscript explored the role of social deprivation to clinical outcome in a cohort of patient in SC who received treatment for prostate cancer. There is a growing interest to elucidate factors other than biology to explain discrepancies in cancer outcome in minority patients and I applause the author's efforts to investigate this importance clinical question. Several comments as follows

1. Page 3 under methods, the authors categorized SDI based on 25th, 50th and 75th percentiles. By doing so SDI is divided to 4 groups - how is this corresponded the low, intermediate and high group as mentioned in the text (only 3 groups)

2. The authors used a p<0.2 as the threshold to decide if a variable should be included into the regression model. The author did provide a reference. However please provide a short justification in the text so that the reads won't need to read another paper to understand the rationale.

3. Table 2 - per text this is supposed to be binary analysis with SDI as high vs low but the table clearly included SDI as three levels. Please clarify. Also please clarify "bivariate analysis" in the title

4. Table 3 described regression model including variables with p< 0.2 from table 2. The p value for Gleason score was 0.1 and therefore should be included.

5. In abstract conclusion the authors mentioned "...neighborhood-level factors ... to improve the precision of cancer control strategies such as operational decisions...health promotion and disease prevention". This is not supported by data presented in the manuscript because the data did not show any associated between SDI and prostate cancer outcome.

6. Need to include author affiliations in the manuscript.

Author Response

Dear Reviewers,

Thank you for your suggestions. Below are point-by-point responses. The changes made in the manuscript are highlighted in red fonts.

Thank you,

Corresponding author.

Reviewer 1

Comments and Suggestions for Authors

The manuscript explored the role of social deprivation to clinical outcome in a cohort of patient in SC who received treatment for prostate cancer. There is a growing interest to elucidate factors other than biology to explain discrepancies in cancer outcome in minority patients and I applause the author's efforts to investigate this importance clinical question.

Several comments as follows

  1. Page 3 under methods, the authors categorized SDI based on 25th, 50th and 75th percentiles. By doing so SDI is divided to 4 groups - how is this corresponded the low, intermediate and high group as mentioned in the text (only 3 groups)

Thank you for your comments. The 25th and 75th percentiles were utilized to categorize to 3 (not 4) groups. The 50th percentile was not utilized. This error has been corrected.

  1. The authors used a p<0.2 as the threshold to decide if a variable should be included into the regression model. The author did provide a reference. However please provide a short justification in the text so that the reads won't need to read another paper to understand the rationale.

Covariates with a P<.20 that were added to the model is justified by previous study where this selection method helped to select the variables to fit the best model in subsequent multivariable logistic regression model.

  1. Table 2 - per text this is supposed to be binary analysis with SDI as high vs low but the table clearly included SDI as three levels. Please clarify. Also please clarify "bivariate analysis" in the title

Thank you for this comment. The “bivariate” was an initial analysis that should have been deleted after newer analysis was performed. “Bivariate” has been deleted and the title has been rewritten as Table 2: Association between Sociodemographic and Clinical Characteristics and Social Deprivation (Chi Square Analysis)

  1. Table 3 described regression model including variables with p< 0.2 from table 2. The p value for Gleason score was 0.1 and therefore should be included.

Thank you for this comment. Gleason score has been included in the model and the newer estimates updated.

  1. In abstract conclusion the authors mentioned "...neighborhood-level factors ... to improve the precision of cancer control strategies such as operational decisions...health promotion and disease prevention". This is not supported by data presented in the manuscript because the data did not show any associated between SDI and prostate cancer outcome.

This statement has been deleted and replaced with “Using SDI to understand the social environment could be particularly useful as part of precision medicine and precision public health approaches and could be used by 2

cancer centers, public health providers, and other health care specialists to inform operational decisions about how to target health promotion and disease prevention efforts in catchment areas and patient populations.”

  1. Need to include author affiliations in the manuscript.

Author affiliations have been provided on Page 1 of the manuscript (and posted below).

1Department of Psychiatry and Behavioral Sciences, Medical University of South Carolina, Charleston, SC 29425.

2Hollings Cancer Center, Medical University of South Carolina, Charleston, SC 29425.

3Department of Public Health Sciences, Medical University of South Carolina, Charleston, SC 29425.

4Department of Cell and Molecular Pharmacology and Experimental Therapeutic Medical University of South Carolina Charleston, SC 29425.

5Department of Behavioral Sciences and Health Education, Rollins School of Public Health, Emory University.

7Department of Medicine, Medical University of South Carolina, Charleston, SC 29425.

8Department of Urology, Medical University of South Carolina, Charleston, SC 29425.

9Department of Population and Public Health Sciences, University of Southern California, Los Angeles CA 90032.

Reviewer 2 Report (New Reviewer)

This study aims to analyze the association of factors (e.g., Gleason score, and marital status)  pertaining to prostate cancer survivors with social deprivation. The presented study can provide an insight for prostate cancer outcomes in terms of improving the treatment outcomes as well as reducing disparities among  a racially diverse cancer population. The paper is well-written and is well-presented.
The following comments need to be addressed:

-For the collected data, need to discuss if the cohort had other comorbidities.
-Including the name of functions used in SAS to carry out
the statistical analysis
-Providing contingency tables for carrying out chi-squared test (to be included in Supplementary Materials)
-Can you provide a plot showing number of prostate cancer survivors from 2011 to 2019?
-Pointing out future direction to the work

Round 2

Reviewer 1 Report (New Reviewer)

The comments have been addressed

Reviewer 2 Report (New Reviewer)

Comments raised in a previous round of review has been addressed. Need to fix the following typo:
*In Line 108, CHANGE
had to 0. 1, 2, 3, and 4.
TO
had to 0, 1, 2, 3, and 4.

This manuscript is a resubmission of an earlier submission. The following is a list of the peer review reports and author responses from that submission.

Round 1

Reviewer 1 Report

Abstract

Overall, the abstract needs work because some important details are missed, the analyses seem quite simple, and I could not understand what was accomplished with the study. The racial distribution was limited to two races as well as the SDI status. Thus, the conclusions are weak and need to be improved. 

Please include the sample size here “A retrospective cohort of prostate cancer patients

Social Deprivation Index (SDI) ” define the scale for a better understanding of the results. Besides, provide the categories for this measure: “high, socially-deprived, etc.”

healthcare professional (HPS) shortage status” this is not defined in the methods.

Results: Approximately 49% ” first, define the population characteristics. Provide the number or percentage of the race of men and the general characteristics of the population assessed. There were only African American and white men?

The mean for SDI was 44.5 (+27.4) and the range was 97 (1-98) ” for men of which race does this correspond?

This study demonstrates that SDI varies considerably among men with prostate cancer ” I was unable to read this considerable variation.

Provide some examples for this: “Greater efforts may be needed to address neighborhood-level factors among African  American men

Introduction

This section is extensive and lacks a clear and brief definition of the context. Overall, the introduction section is devoid of a clear explanation regarding how precision medicine is enhanced by knowing socioeconomic features of a population affected by a disease. This is also the case for the present study because I was unable to understand how describing these factors should improve the health of PC survivors. Authors should elaborate more on this, avoid extensive citation of literature and reduce the length of this section.   

The first paragraph should be either removed because does not help understanding the importance of the study. 

Just as decisions about early detection occur within a community and clinical context, diseases such as cancer occur within the general social context of an individual’s life and community.” This is not clear, please rewrite o remove this.

The third paragraph seems more like a philosophical dissertation than an effort to establish the context of a scientific study. Authors are encouraged to reduce to one third this excessive use of sentences, references, and ideas that do not contribute to the study. For instance: “About 27% of the population and 67% of counties in South Carolina are designated as rural. ("Bunch B. Department of Commerce. South Carolina Division of Research. Available ahttps://dc.statelibrary.sc.gov/bitstream/handle/10827/15126/DOC_Analy- 90 sis_of_Rural_Definition_2008-1.pdf. Downloaded on 04/12/2020.," ; Weis et al., 2010; 91 Weissman et al., 2014) 

Furthermore, many South Carolina residents have low education and incomes, and about 12.7 % of residents do not have health insurance” Again, what is the purpose of including this information? Even though the study includes a sample size from this state, the overall context should not focus exclusively on describing the characteristics of this state, instead the authors should present a general context in which their study fits and further allows the generalization of their findings.   

Please change the format of the reference to avoid including the full bibliographic information.

The last paragraph pf the intro is the more informative, though also needs to be rewritten. For instance, “Knowing social deprivation is important for precision medicine approaches because 115 it provides insight about neighborhood level factors that could be targeted as part of can- 116 cer control interventions. ” this is part of the rationale for performing the study. However, the study focuses on cancer survivors that were treated with radical prostatectomy, and the introduction mentions control interventions. Thus, I fell like there rationale does not include this particular population “cancer survivors” I recommend that authors should rewrite the full introduction and they should improve the rationale for their study.

based on their clinical characteristics (e.g., Gleason score, tumor stage) ” the results presented in the abstract lacked this important clinical characteristics. 

Methods

Overall,

Healthcare professional shortage status (SD) ” this variable is not defined in the methods section.

Multivariate logistic regression analysis was conducted to identify ” the abstract only says “logistic regression” Please include more details regarding the modelling approach. 

Results

Overall, the results section seems weak and simple.

Table 1. “Healthcare professional shortage status (SD) ” is repeated as a row in the first column. The correct entrance should omit the SD.

Table 2 presents data only for %higher deprivation and no info is presented for lower SDI. Thus I could not understand how did authors performed a Chi square based only on the higher deprivation category. “Table 2 shows the bivariate analysis of social deprivation using the binary SDI 177 variable of high versus low categories 

Figure 1 is of low quality for a study like this, please correct the legends for the figure avoiding the grey filling color. Besides, what is the purpose of presenting SDI by county given that this geographical level was not used during the analyses? 

Author Response

Reviewer 1

 Abstract

 Overall, the abstract needs work because some important details are missed, the analyses seem quite simple, and I could not understand what was accomplished with the study. The racial distribution was limited to two races as well as the SDI status. Thus, the conclusions are weak and need to be improved. 

Important details have been added to the abstract as follows:

-# 2 below: A retrospective cohort of 253 prostate cancer patients (added number “253” which is the sample size for this study)

-Of the 253 participants in our sample, 168 were Whites, 81 were African Americans, 1 was Hispanic and 3 were more than 1 race. The study excluded the 1 Hispanic and 3 that identified as multiple races because adding these patients in will result in very small groups that will reduce the power of any conclusions. This new sentence was included in the Methods Section of the main manuscripts and not in the abstract because of word count limitations.

  1. Please include the sample size here “A retrospective cohort of prostate cancer patients…”

“Social Deprivation Index (SDI) ” define the scale for a better understanding of the results. Besides, provide the categories for this measure: “high, socially-deprived, etc.”

Edits made as follows:

-A retrospective cohort of 253 prostate cancer patients (added number “253” which is the sample size for this study)

-The SDI variable were defined in the Methods Section as written in the next sentence. “The 25th, 50th and 75th percentile were utilized to categorize the SDI into lower deprivation, intermediate deprivation and higher deprivation areas. The median value was used to categorize patients into groups who were living in geographic areas with high versus low levels of social deprivation”. The definition of SDI was not included in the abstract because of word count limitation.

  1. “healthcare professional (HPS) shortage status” this is not defined in the methods.

-This has been defined in the methods.

-County-level Healthcare Professional Shortage (HPSA) status was extracted from Health Resources and Services Administration. HPSA values ranged from 1 to 26 with higher values representing greater health care professional shortage. The 25th, 50th, and 75th percentile value were used to categorize patients into groups who were living in geographic areas with high versus low levels of health care professional shortage. Lower shortage (better) was categorized as HPSA scores 8–10, intermediate shortage was categorized as HPSA scores of 11-15, while higher deprivation (worse) was categorized as SDI scores 16–20.

  1. “Results: Approximately 49% ” first, define the population characteristics.

-the following sentence was added prior to the referenced sentence above. It now reads as “The cohort of 253 men consisted of 168 White, 81 African American, 1 Hispanic and 3 multiracial men. Approximately 49% of 249 men lived in areas with high SDI…”.

  1. Provide the number or percentage of the race of men and the general characteristics of the population assessed. There were only African American and white men?

-please see item #4 above.

  1. “The mean for SDI was 44.5 (+27.4) and the range was 97 (1-98) ” for men of which race does this correspond?

The clause “for all study participants” was added to clarify that this value represents all study participants because this was for all 249participants.

  1. “This study demonstrates that SDI varies considerably among men with prostate cancer ” I was unable to read this considerable variation.

-The sentence was rewritten as follows: This study demonstrates that SDI varies considerably “by race” among men with prostate cancer treated with radical prostatectomy.       

  1. Provide some examples for this: “Greater efforts may be needed to address neighborhood-level factors among African  American men…”

-an example was provided so the sentence now reads as “Greater efforts may be needed to address neighborhood-level factors among African American men who have a personal history of prostate cancer to improve the precision of cancer control strategies such as operational decisions about how to target health promotion and disease prevention efforts in catchment areas and patient populations”.

Introduction

  1. This section is extensive and lacks a clear and brief definition of the context. Overall, the introduction section is devoid of a clear explanation regarding how precision medicine is enhanced by knowing socioeconomic features of a population affected by a disease. This is also the case for the present study because I was unable to understand how describing these factors should improve the health of PC survivors. Authors should elaborate more on this, avoid extensive citation of literature and reduce the length of this section.   

The length has been reduced. The first paragraph was removed as suggested (item #10 below). Additional sentence was removed as suggested (item # 11 below). To provide a clear explanation regarding how precision medicine is enhanced by knowing socioeconomic features of a population affected by a disease, we included the following sentence “According to the definition by the National Institute of Health (NIH), precision medicine refers to a new treatment and prevention method based on understanding of individual gene, environment, and lifestyle”. This study provides an understanding of the “environment” part of precision medicine.

  1. The first paragraph should be either removed because does not help understanding the importance of the study. 

The first paragraph was removed as suggested.

  1. “Just as decisions about early detection occur within a community and clinical context, diseases such as cancer occur within the general social context of an individual’s life and community.” This is not clear, please rewrite o remove this.

This sentence was removed as suggested.

  1. The third paragraph seems more like a philosophical dissertation than an effort to establish the context of a scientific study. Authors are encouraged to reduce to one third this excessive use of sentences, references, and ideas that do not contribute to the study. For instance: “About 27% of the population and 67% of counties in South Carolina are designated as rural. ("Bunch B. Department of Commerce. South Carolina Division of Research. Available ahttps://dc.statelibrary.sc.gov/bitstream/handle/10827/15126/DOC_Analy- 90 sis_of_Rural_Definition_2008-1.pdf. Downloaded on 04/12/2020.," ; Weis et al., 2010; 91 Weissman et al., 2014) ”.

Paragraph reduced by removing sentences as suggested. Format of references also changed.

  1. “Furthermore, many South Carolina residents have low education and incomes, and about 12.7 % of residents do not have health insurance” Again, what is the purpose of including this information? Even though the study includes a sample size from this state, the overall context should not focus exclusively on describing the characteristics of this state, instead the authors should present a general context in which their study fits and further allows the generalization of their findings.   

This sentence was removed as suggested.

  1. Please change the format of the reference to avoid including the full bibliographic information.

Format of references changed as suggested.

  1. The last paragraph of the intro is the more informative, though also needs to be rewritten. For instance, “Knowing social deprivation is important for precision medicine approaches because 115 it provides insight about neighborhood level factors that could be targeted as part of cancer control interventions. 116 ” this is part of the rationale for performing the study. However, the study focuses on cancer survivors that were treated with radical prostatectomy, and the introduction mentions control interventions. Thus, I fell like there rationale does not include this particular population “cancer survivors” I recommend that authors should rewrite the full introduction and they should improve the rationale for their study.

The last paragraph has been rewritten by adding “the following sentence “According to the definition by the National Institute of Health (NIH), precision medicine refers to a new treatment and prevention method based on understanding of individual gene, environment, and lifestyle”. This study provides an understanding of the “environment” part of precision medicine. We also added “Geographic variables are important in cancer outcomes among cancer survivors as geographic inequalities indicate a potential for improving cancer care and survival”.

  1. “based on their clinical characteristics (e.g., Gleason score, tumor stage)” the results presented in the abstract lacked these important clinical characteristics. 

-a sentence was added to the abstract as follows: “There were no significant association between SDI and clinical characteristics of prostate cancer”.

Methods

  1. Overall, “Healthcare professional shortage status (SD) ” this variable is not defined in the methods section. “Multivariate logistic regression analysis was conducted to identify ” the abstract only says “logistic regression” Please include more details regarding the modelling approach. 

-“Healthcare professional shortage status has been defined as follows: “County-level Healthcare Professional Shortage (HPSA) status was extracted from Health Resources and Services Administration. HPSA values ranged from 1 to 26 with higher values representing greater health care professional shortage. The 25th, 50th, and 75th percentile value were used to categorize patients into groups who were living in geographic areas with high versus low levels of health care professional shortage. Lower shortage (better) was categorized as HPSA scores 8–10, intermediate shortage was categorized as HPSA scores of 11-15, while higher deprivation (worse) was categorized as SDI scores 16–20”.

-“ multivariable” added in the abstract as “multivariable” logistic regression

-Additional information provided about multivariable logistic regression as follows: “Multivariate logistic regression analysis was conducted to identify factors having significant independent associations with SDI. Variables that had a p<0.20 with SDI in the bivariate analysis was included in the regression model. In exploring the multivariable logistic regression determining SDI, the variables that were entered into the model are race, age, marital status and HPSA.  In exploring the multivariable logistic regression between race, SDI and biochemical reoccurrence, the variables that were entered into the model are race and SDI with adjustments for healthcare professional shortage, age, marital status, prostate specific antigen, stage, grade/Gleason score.”.

Results

Overall, the results section seems weak and simple.

  1. Table 1. “Healthcare professional shortage status (SD) ” is repeated as a row in the first column. The correct entrance should omit the SD.

-corrections made

  1. Table 2 presents data only for %higher deprivation and no info is presented for lower SDI. Thus I could not understand how did authors performed a Chi square based only on the higher deprivation category. “Table 2 shows the bivariate analysis of social deprivation using the binary SDI 177 variable of high versus low categories ”

-corrections made, please see Table 2 which now contains 3 categories instead of 1. Also because of the comments in item # 23 (below) by 2nd reviewer, SDI was grouped into 3 instead of into 2 categories as suggested.

  1. Figure 1 is of low quality for a study like this, please correct the legends for the figure avoiding the grey filling color. Besides, what is the purpose of presenting SDI by county given that this geographical level was not used during the analyses? 

Figure 1 has been deleted from the study as suggested.

Reviewer 2

  1. The topic of prostate cancer is important for obvious reasons, most notably because it is the most common form of cancer among men (besides skin cancer).  However, the present study is not very informative.  All it tells us is that, in a sample of prostate cancer survivors, African American men are more likely than are White men to live in residential areas that are socioeconomically disadvantaged and medically underserved.  This is, of course, the same pattern of racial inequality found in the general population and, hence, we learn nothing new from the authors' results. The study might produce some interesting findings if the authors analyzed a sample of radical prostatectomy (RP) patients who experienced biochemical relapse (say after five or 10 years), controlling for Gleason score and other prognostic indicators.

-Biochemical reoccurrence data has been added as Table 4.

- African Americans had a higher likelihood of developing biochemical reoccurrence (OR=3.7, 95% C.I.-1.7-8.0) compared with White men after adjustments were made for healthcare professional shortage, age, marital status, prostate specific antigen, stage, grade/Gleason score  (Table 4). There were no statistically significant associations between SDI and biochemical reoccurrence.  

  1. There are other problems, too.  The formatting of much of the paper's introductory section is really messed-up, especially in lines 89-114.  The authors' methodological decisions are not explained or justified with citations to other studies. 

-formatting issues corrected

-methodological justification provided e.g. 1, Covariates with a P<.20 that were added to the model is justified by previous study. (<references provided>) E.g., 2, Biochemical reoccurrence (yes/no) was determined by prostate specific antigen value > 0.2 ng/mL. (<references provided>)

  1. I don't understand why the continuous variables, SDI and Healthcare Professional Shortage Status are categorized as "high" or "low."  Why not simply analyze these variables on a continuous scale?  Alternatively, the authors might use "high," "medium" or "low" categories as the measurement scale for these variables.  High versus low is just too crude and simplistic.

-SDI and HPSA grouped to 3 as suggested.

-The SDI variable were defined in the Methods Section as written in the next sentence. “The 25th, 50th and 75th percentile were utilized to categorize the SDI into lower deprivation, intermediate deprivation and higher deprivation areas. The median value was used to categorize patients into groups who were living in geographic areas with high versus low levels of social deprivation”. The definition of SDI was not included in the abstract because of word count limitation.

-County-level Healthcare Professional Shortage (HPSA) status was extracted from Health Resources and Services Administration. HPSA values ranged from 1 to 26 with higher values representing greater health care professional shortage. The 25th, 50th, and 75th percentile value were used to categorize patients into groups who were living in geographic areas with high versus low levels of health care professional shortage. Lower shortage (better) was categorized as HPSA scores 8–10, intermediate shortage was categorized as HPSA scores of 11-15, while higher deprivation (worse) was categorized as SDI scores 16–20.

  1. Finally, why is the sample limited to those men who have been treated surgically (RP)?  Why not include those who have been treated with radiation (external beam and/or seeds) or other methods (e.g., cryosurgery or HIFU)?

-One limitation of this study is that most patients at our institution were treated with radical prostatectomies; adding many types of radiation therapies e.g., brachytherapy, photon beams, Image-Guided, Intensity-Modulated Radiation Therapy, or Image-Guided Radiation Therapy would result in very small treatment groups that will reduce the power of any conclusions.

-the statement above has been added to limitations of the study.

Reviewer 2 Report

The topic of prostate cancer is important for obvious reasons, most notably because it is the most common form of cancer among men (besides skin cancer).  However, the present study is not very informative.  All it tells us is that, in a sample of prostate cancer survivors, African American men are more likely than are White men to live in residential areas that are socioeconomically disadvantaged and medically underserved.  This is, of course, the same pattern of racial inequality found in the general population and, hence, we learn nothing new from the authors' results.

The study might produce some interesting findings if the authors analyzed a sample of radical prostatectomy (RP) patients who experienced biochemical relapse (say after five or 10 years), controlling for Gleason score and other prognostic indicators.

There are other problems, too.  The formatting of much of the paper's introductory section is really messed-up, especially in lines 89-114.  The authors' methodological decisions are not explained or justified with citations to other studies.  I don't understand why the continuous variables, SDI and Healthcare Professional Shortage Status are categorized as "high" or "low."  Why not simply analyze these variables on a continuous scale?  Alternatively, the authors might use "high," "medium" or "low" categories as the measurement scale for these variables.  High versus low is just too crude and simplistic.

Finally, why is the sample limited to those men who have been treated surgically (RP)?  Why not include those who have been treated with radiation (external beam and/or seeds) or other methods (e.g., cryosurgery or HIFU)?

Author Response

(The authors gave the same response as above.)

Round 2

Reviewer 1 Report

The authors addressed al the queries and comments.

Reviewer 2 Report

I'm just not convinced that the paper makes a significant contribution to the published literature.